Changes of diazotrophic communities in response to cropping systems in a Mollisol of Northeast China

Zou Jiaxun 1 2
http://orcid.org/0000-0001-6617-2727 Yao Qin 1
Liu Junjie 1
Li Yansheng 1
Song Fuqiang 2
Liu Xiaobing 1
Wang Guanghua 1 wanggh@iga.ac.cn
1 Key Laboratory of Mollisols Agroecology, Northeast Institute of Geography and Agroecology, Chinese Academy of Sciences , Harbin , China
2 College of Life Science, Heilongjiang University , Harbin , China
Landa Blanca
Electronic publication date: 2020 Jul 15
Publication date: 2020
Volume: 8
Electronic Location ID: e9550
Received 2020 Jan 6; Accepted 2020 Jun 24
Copyright: © 2020 Zou et al.
Copyright year: 2020
Copyright holder: Zou et al.
License: This is an open access article distributed under the terms of the Creative Commons Attribution License, which permits unrestricted use, distribution, reproduction and adaptation in any medium and for any purpose provided that it is properly attributed. For attribution, the original author(s), title, publication source (PeerJ) and either DOI or URL of the article must be cited.
License URL: https://creativecommons.org/licenses/by/4.0/

Keywords: Black soil, Bradyrhizobium sp., Continuous cropping, Crop rotation, Diversity, nifH gene

Funding: National Key Research and Development Program of China 2017YFD0200604 This work was supported by grants from National Key Research and Development Program of China (2017YFD0200604). The funders had no role in study design, data collection and analysis, decision to publish, or preparation of the manuscript.

==============================
Nitrogen-fixing microorganisms play important roles in N cycling. However, knowledge related to the changes in the diazotrophic community in response to cropping systems is still rudimentary. In this study, the nifH gene was used to reveal the abundance and community compositions of diazotrophs in the cropping systems of continuous cropping of corn (CC) and soybean (SS) and soybean-corn rotation for growing corn (CSC) and soybean (SCS) in a black soil of Northeast China. The results showed that the abundance of the nifH gene was significantly higher in cropping soybean than in cropping corn under the same cropping system, while remarkably increased in the rotation system under the same crop. The Shannon index in the CC treatment was significantly higher than that in the other treatments, but the OTU number and Chao1 index had no significant change among the four treatments. Bradyrhizobium japonicum was the dominant diazotrophic species, and its relative abundance was at the lowest value in the CC treatment. In contrast, Skermanella sp. had the highest relative abundance in the CC treatment. A PCoA showed that the diazotrophic communities were separated between different cropping systems, and the variation caused by continuous corn cropping was the largest. Among the tested soil properties, the soil available phosphorus was a primary factor in determining diazotrophic community compositions. Overall, the findings of this study highlighted that the diazotrophic communities in black soils are very sensitive to cropping systems.

Introduction

The black soil zone of Northeast China is one of the four large Mollisol regions in the world and plays an important role in maintaining food security in China (Liu et al., 2008). In this region, soybean and corn are two major crops that are either continuously cropped or grown annually in rotation with each other. It is well known that continuous cropping can decline soil quality, which seriously decreases the quantity and quality of crop products. Previous studies have shown that continuous cropping has led to several problems in soils, for example, deficiencies in soil nutrition (Ashworth et al., 2018), decreases in soil enzyme activity (Chavarría et al., 2016), increases in the autotoxicity of root exudates (Huang et al., 2013), increases in pests and diseases (Torres et al., 2018) and imbalances in soil microbial communities (Bennett et al., 2012). Although the problems caused by continuous cropping of corn is less harmful than that of continuous cropping of soybean (Xu, Li & Li, 2004), several studies have revealed that continuous cropping of corn has caused the lodging and surge of pests, which have damaged corn yield (Jirak-Peterson & Esker, 2011; Liang et al., 2017). The cause of the barriers of continuous cropping are very complex, and some biotic and abiotic factors are commonly related to crop-yielding decline. Among these factors, biotic factors, such as changes in soil microbial communities, are often considered to be the major reason for barriers to continuous cropping (Dias, Dukes & Antunes, 2015). Several studies have shown that continuous cropping destroyed the intrinsic balance of soil microorganisms and increased abundance of crop pathogens such as soil-borne Fusarium spp. (Xiong et al., 2015; Zhu et al., 2018).

Biological nitrogen fixation (BNF) is the main route of inputting nitrogen (N) in natural ecosystems. Approximately 52–130 Tg of N is input into terrestrial ecosystems annually through the BNF method (Davies-Barnard & Friedlingstein, 2020). BNF is carried out by a wide range of microorganisms containing nitrogen-fixing enzymes. The enzymes are regulated by multiple genes, and the nifH gene is one of the most conserved functional genes and is widely used as a biomarker gene for studying nitrogen-fixing microorganisms or diazotrophic communities in soils (Coelho et al., 2009). A plethora of studies have shown that the diazotrophic community is highly sensitive to variations in soil factors such as pH (Fan et al., 2018), organic matter (Calderoli et al., 2017) and available nutrient content (Collavino et al., 2014). Since soil factors are greatly affected by different cropping systems (Jha, Prasad & Misra, 2004; Moisander et al., 2012), whether different cropping systems have a direct or indirect impact on diazotrophic communities in soils, especially in black soils, is rarely reported.

Using molecular fingerprinting methods, such as polymerase chain reaction-denaturing gradient gel electrophoresis (PCR-DGGE) and the terminal restriction fragment length polymorphism (T-RFLP) method, several studies have investigated the changes in diazotrophic communities in response to cropping systems. For example, Pereira E Silva et al. (2013) revealed that NH4+-N, NO3−-N, and soil pH were the determining factors in shifting the diazotrophic community structure by using the PCR-DGGE method. Using the T-RFLP method, Xiao et al. (2010) revealed that diazotrophic community diversity in continuous cropping of soybean fields was different from that in rotational cropping. However, the low resolutions of molecular fingerprinting methods restricted in-depth data analysis and did not allow us to fully understand the changes in diazotrophic community structures. Recently, high-throughput sequencing (HTS) has become the mainstream method for studying diazotrophic communities (Wang et al., 2017a; Hu et al., 2018). Using this method, Wang et al. (2017b) revealed that soil pH and nutrient availability had a cooperative effect on diazotroph abundance, while soil nutrient availability was the main factor. Noticeably, most studies of using HTS method to analyze the changes in microbial community structures in response to different cropping systems have mainly focused on bacterial and fungal communities (Bai et al., 2015; Liu et al., 2017), rarely specifically related to the diazotrophic community.

In this study, we comparatively investigated the abundance, diversity and community structures of diazotrophs under different cropping systems in the black soil region of Northeast China using real-time PCR and Illumina MiSeq sequencing methods. Specifically, the purposes of this study were (1) to compare the effect of different cropping systems on soil chemical properties; (2) to assess the responses of diazotrophic abundance and diversity to different cropping systems; and (3) to examine the contributions of crop types and soil factors to the changes of diazotrophic community structures.

Materials and Methods

Study site descriptions and soil sampling

A fixed experimental field was established in 2013 at Guangrong Village (47°21′N, 126°49′E), Hailun County, Heilongjiang Province of Northeast China, to evaluate the long-term effectiveness of crop rotation and continuous cropping systems. At the experimental site, the annual mean temperature and precipitation were 1.5 °C and 530 mm, respectively. The soil in the experimental site was a typical black soil, which classified as Mollisol according to the USA’s soil taxonomy.

Four cropping treatments were evaluated in this study: continuously cropped corn (CC), continuously cropped soybean (SS), soybean-corn rotation for growing corn crops (CSC) and soybean-corn rotation for growing soybean crops (SCS) at the study year. The cropping sequences for the four treatments from the starting year of 2013 to the sampling year of 2017 is shown in Fig. S1. Commercial fertilizers urea, diammonium phosphate and potassium sulfate were used as N, P and K sources. The dosage of chemical fertilizers used for growing soybean and corn was based on the local farming practice. For cropping soybean, chemical fertilizers were applied before sowing at the rate of 35.2 kg P ha−1, 55.2 kg N ha−1 and 22.4 kg K ha−1. For cropping corn, chemical fertilizers were supplied before sowing as base fertilizers at the rate of 35.2 kg P ha−1, 66.9 kg N ha−1 and 22.4 kg K ha−1. Additionally, 105 kg N ha−1 was supplied as top dressing at the V6 stage (6th leaf collars visible) of corn.

Each treatment had four randomly arranged replicate plots, and each plot was 75.6 m2 containing 12 rows with a row length of 9 m and width of 70 cm. Soil samples were collected on 14 September 2017 from the field when soybean was nearly at maturity (R7 stage) and corn was at wax maturity. From each plot, five points of soil within a depth of 0–20 cm were randomly collected and combined into a single sample. The soil was placed into individual sterile plastic bags, which were placed in ice boxes and transferred back to the laboratory immediately. At the laboratory, the soil was sieved through a 2-mm mesh and divided into two parts: one part was stored at −80 °C for DNA extraction, and the remaining soil was kept at 4 °C for soil chemical analysis.

Soil chemical property analysis

Soil pH was determined using a pH meter in the soil-water suspension (1:2.5 w/v). The concentrations of soil total carbon (TC) and total nitrogen (TN) were determined using an elemental analyzer (VarioEL III, Germany). Soil NH4+-N and NO3−-N extracted with 2.0 M KCl solution, total phosphorus (TP) digested by HClO4-H2SO4 and available phosphorus (AP) extracted with NaHCO3 were measured with a continuous flow analytical system (SKALAR SAN++, The Netherlands). Soil total potassium (TK) digested with HNO3-HClO4-HF and available potassium (AK) extracted with 1.0 M NH4Ac were estimated using inductively coupled plasma-atomic emission spectrometry (ICPS-7500, Shimadzu, Japan) (Lu, 2000).

Soil DNA extractions and quantification of nifH gene abundance

The total soil DNA of each sample was extracted from 0.5 g of soil samples kept in −80 °C freezer using a FastDNA® SPIN Kit for Soil (MP Biomedicals, Santa Ana, CA, USA) following the manufacturer’s protocols. The extracted DNA was diluted in DES buffer (DNA Elution Solution-Ultra Pure Water). DNA concentration was determined with the NanoDrop method (NanoDrop 2000; Thermo Scientific, Waltham, MA, USA) and then stored at −20 °C until further use.

Quantitative real-time PCR (qPCR) was performed using the primers nifH-F (5′-AAA GGY GGW ATC GGY AAR TCCA CCA C-3′) and nifH-R (5′-TTG TTS GCS GCR TAC ATS GCC ATC AT-3′) to measure the abundance of the nifH gene in a LightCycler®480 (Roche Applied Science, Basel, Switzerland). Each qPCR reaction mixture contained 10 μl of SYBR Premix Ex TaqTM (Takara, Dalian, China), 1 μl of extracted soil DNA, 1 μl of 10 μm forward primer, 1 μl of 10 μm reverse primer and 7.0 μl of sterilized MilliQ water (Rösch, Mergel & Bothe, 2002). For each sample, qPCR amplification was performed in triplicate following a program of denaturation at 95 °C for 30 s (ramp rate of 4.4 °C/s), 30 amplification cycles of 95 °C for 5 s, 60 °C for 30 s and 50 °C for 30 s. The nifH gene copy numbers were calculated using a regression equation for converting the cycle threshold (Ct) value to the known number of cloned nifH gene ranged from 5.07 × 102 to 5.07 × 108 gene copies per μL in the standards (Hu et al., 2018).

Illumina MiSeq sequencing

The diazotrophic community was analyzed using the nifH gene-specific primers nifH-F/nifH-R with 7-bp unique barcodes at the 5′ end, which produced an approximately 432 bp PCR product. The PCRs were performed in triplicate for each sample in ABI GeneAmp® 9700 with a volume of 20 μl, which comprised 4 μl of 5 × FastPfu Buffer, 2 μl of 2.5 mm dNTPs, 0.8 μl (5 μm) of each primer, 0.4 μl of FastPfu Polymerase, 0.2 μl of BSA (Bovine Serum Albumin) solution, 1.0 μl of template DNA (10 ng) and ddH2O to reach 20 μl. The PCR amplification program started with 95 °C for 3 min for initial denaturation, followed by 35 cycles of 95 °C for 30 s for denaturation, 55 °C for 30 s for annealing, 72 °C for 45 s, and ended with one final cycle at 72 °C for 10 min for extension. Equimolar amounts of the purified PCR products were pooled and paired-end sequenced on an Illumina MiSeq platform at Majorbio BioPharm Technology Co., Ltd., Shanghai, China.

Sequence data analysis

After sequencing, QIIME Pipeline (Version 1.9.0) was used to process the raw data (http://qiime.org/tutorials/tutorial.html) (Caporaso, Kuczynski & Stombaugh, 2010). Sequences shorter than 200 bp or with an average base quality score below 20 were removed before further analysis. Removal of the trimmed sequences was done with the UCHIME algorithm (Edgar et al., 2011). Clustering of the high-quality sequences into operational taxonomic units (OTUs) was performed through the UPARSE software (http://drive5.com/uparse/) at a 97% similarity level. Representative sequences from each OTU were translated into amino acid sequences and taxonomic identification was performed using a BLASTp search tool at the NCBI website (https://blast.ncbi.nlm.nih.gov/). Visualization of microbial communities at the OTU level in different samples was conducted using Circos (Krzywinski et al., 2009). All raw sequences generated in this study have been deposited in NCBI under the accession PRJNA516581.

Statistical analysis

To compare the difference between samples, the lowest sequence number of 6,545 was randomly selected from each sample as subset data for diazotrophic diversity analyses. The alpha diversity of diazotrophic community was presented by Shannon, Simpson, OTU number and Chao1 indices. SPSS software (Version 22.0) was used to evaluate the significant differences in soil properties, diazotrophic abundance and alpha diversity between treatments by one-way ANOVA analysis, as well as to calculate the correlations between the community compositions of diazotroph and soil properties. Based on a UniFrac distance matrix, the beta diversity of diazotrophic community was analyzed through principal coordinate analysis (PCoA) in the R environment (version 3.2.5) (R Development Core Team, 2016). The difference in the diazotrophic community between treatments was analyzed using the Adonis method based on Bray–Curtis distances (Clarke, 1993), which was conducted using the “vegan” library in the R environment. The relationships between the soil properties and composition of the diazotrophic community were analyzed by Redundancy Analysis (RDA) and envfit analysis in the R environment using the “vegan” library.

Results

Changes of soil properties

The effects of different cropping systems on soil chemical properties are summarized in Table 1. The soil pH, TK, AK and TC contents were not significantly changed among the treatments. The AP content varied significantly among treatments (P < 0.05), with the highest and the lowest values in the SCS and CC treatments, respectively. C/N was significantly higher in the CC and SS treatments than in the CSC and SCS treatments (P < 0.05). The contents of soil TN, NH4+-N and NO3−-N were significantly changed among the treatments (P < 0.05), with the highest values in the CSC treatment. Overall, this result indicated that different cropping systems significantly changed the soil nutrient content, especially the chemical properties related to nitrogen nutrition.

Table 1 Effects of different cropping systems on soil chemical properties.

Treatment	pH	TN (g kg−1)‡	TC (g kg−1)‡	C/N	TP (g kg−1)‡	TK (g kg−1)‡	NH4+-N (mg kg−1)	NO3--N (mg kg−1)	AP (mg kg−1)‡	AK (mg kg−1)‡	
CC†	6.39 ± 0.12a§	1.57 ± 0.16bc	21.94 ± 2.62a	13.99 ± 0.50a	0.56 ± 0.029a	15.64 ± 0.25a	18.83 ± 1.37b	30.66 ± 5.21b	15.98 ± 1.62c	231.05 ± 14.11a	
SS†	6.36 ± 0.01a	1.52 ± 0.12c	20.90 ± 1.88a	13.73 ± 0.17a	0.56 ± 0.035a	15.79 ± 0.20a	17.74 ± 1.68b	13.69 ± 3.64c	20.00 ± 2.08b	219.55 ± 11.29a	
CSC†	6.36 ± 0.16a	1.84 ± 0.15a	20.97 ± 2.40a	11.40 ± 0.48b	0.47 ± 0.023b	15.97 ± 0.16a	24.22 ± 1.35a	37.98 ± 3.23a	20.68 ± 2.46b	223.57 ± 9.84a	
SCS†	6.34 ± 0.09a	1.78 ± 0.13ab	20.47 ± 2.67a	11.50 ± 0.86b	0.56 ± 0.035a	15.68 ± 0.40a	18.13 ± 1.12b	10.28 ± 1.42c	25.96 ± 2.40a	234.96 ± 12.21a	
Notes:

† CC, SS, CSC and SCS represent continuous corn cropping, continuous soybean cropping, soybean-corn rotation for growing crop was corn and soybean-corn rotation for growing crop was soybean, respectively.

‡ TN, TC, TP, TK, AP and AK represent soil total nitrogen, total carbon, total phosphorus, total potassium, available phosphorus and available potassium, respectively.

§ Different letters within the same column indicate significant difference between treatments tested by one-way ANOVA (P < 0.05). Values are the means ± SE (n = 4).

Abundance of the nifH gene

The abundance of the nifH gene varied significantly among treatments (P < 0.01), and it ranged from an average of 2.3 × 106 to 11.1 × 106 gene copies per gram of soil under different cropping systems (Fig. 1A). The abundance of the nifH gene in the SCS treatment was 57.4%, 123.8% and 382.3% higher compared to that in SS, CSC and CC, respectively. Pearson’s correlation analysis showed that nifH gene abundance was significantly correlated with C/N (r = −0.541, P = 0.031), NO3−-N (r = −0.602, P = 0.014), and AP (r = 0.597, P = 0.015) (Figs. 1B–1D).

Figure 1 Effect of different cropping systems on diazotrophic nifH gene abundance (A) and the bivariate correlation between the abundance of the nifH gene and soil C/N (B), NO3−-N (C) and AP content (D) in black soil.

CC and SS represent the treatments of continuous cropping corn and soybean, respectively; CSC and SCS represent the treatments of soybean-corn rotation for growing corn and soybean, respectively. Error bars show the standard deviation of abundance, and the bars marked with different letters show the significant difference at P < 0.05.

Alpha diversity index of the diazotrophic community

The alpha diversity of the diazotrophic community showed that the Shannon and Simpson indices had opposite results. CC treatment had the highest score with Shannon index but had the lowest score with Simpson index. The OTU numbers and Chao1 values were not significantly different among treatments (Fig. 2). Pearson’s correlation analysis showed that the Shannon index had a significant positive correlation with C/N (r = 0.517, P = 0.040) and a significant negative correlation with AP (r = −0.664, P = 0.005). In contrast, the Simpson index showed the opposite trend (Table 2).

Figure 2 Effect of different cropping systems on the Shannon indice (A), Simpson indice (B), OTU number (C) and Chao1 indice (D) of alpha diversity of diazotrophic communities calculated based on a randomly selected subset of 6545 sequences per sample.

The abbreviations of CC, SS, CSC and SCS are described in Fig. 1. The bars marked with different letters show the significant difference at P < 0.05.

Table 2 The bivariate correlation between the alpha diversity of diazotrophic communities and soil factors.

Diversity index†	pH	TN‡	TC‡	C/N	NH4+-N	NO3−-N	AP‡	AK‡	TP‡	TK‡	
OTU number	−0.163	−0.095	−0.245	−0.121	0.146	−0.020	−0.297	−0.063	−0.009	−0.048	
Shannon	0.095	−0.329	0.167	0.517*§	0.006	0.356	−0.664**§	0.037	0.107	−0.146	
Simpson	−0.099	0.292	−0.171	−0.478*	−0.032	−0.421	0.627**	−0.096	−0.111	0.161	
Chao1	−0.088	0.253	0.072	−0.174	−0.170	−0.267	−0.013	0.065	0.187	0.117	
Notes:

† All indices are calculated based on the minimum number of 6,545 sequences per sample.

‡ TN, TC, TP, TK, AP and AK represent soil total nitrogen, total carbon, total phosphorus, total potassium, available phosphorus and available potassium, respectively.

§ * and ** represent significant correlation at P < 0.05 and P < 0.01 level, respectively.

Diazotrophic community composition

A total of 205,696 high quality sequences were obtained across all samples, with an average of 12,853 sequences per sample. A random subset of 6,545 (minimum number of sequences) sequences was selected from each sample for downstream analysis. Proteobacteria was the dominant phylum in all treatments, with average relative abundances ranging from 99.91% to 100% across four treatments. Within this phylum, the Alphaproteobacteria was the dominant class, with an average relative abundance ranging from 94.35% to 96.13%, while the highest relative abundances of Betaproteobacteria and Deltaproteobacteria were only 1.37% and 1.71%, respectively. In addition, a very low abundance of Cyanobacteria was detected in some samples (Table 3).

Table 3 Relative abundances of the dominant diazotrophic bacteria at different taxonomic levels in different cropping systems.

Taxa	Treatments	
CC†	SS†	CSC†	SCS†	
Phylum					
Cyanobacteria	0.08 ± 0.01a‡	0.09 ± 0.03a	0.00 ± 0.01b	0.03 ± 0.02a	
Proteobacteria	99.92 ± 0.13b	99.91 ± 0.13b	100.00 ± 0.01a	99.97 ± 0.05b	
Class					
Alphaproteobacteria	94.35 ± 1.87a	95.33 ± 0.55a	94.93 ± 0.83a	96.13 ± 1.45a	
Betaproteobacteria	1.37 ± 0.37a	0.97 ± 0.28ab	1.30 ± 0.16a	0.65 ± 0.30b	
Deltaproteobacteria	1.56 ± 0.93a	1.20 ± 0.48a	1.71 ± 0.84a	0.85 ± 0.20a	
Order					
Burkholderiales	1.47 ± 0.37a	0.98 ± 0.28bc	1.30 ± 0.13ab	0.55 ± 0.32c	
Desulfuromonadales	1.48 ± 0.92a	1.15 ± 0.49a	1.65 ± 0.88a	0.80 ± 0.18a	
Myxococcales	0.16 ± 0.16a	0.05 ± 0.01a	0.06 ± 0.05a	0.05 ± 0.03a	
Nostocales	0.01 ± 0.01a	0.02 ± 0.01a	0.01 ± 0.01a	0.03 ± 0.02a	
Rhizobiales	81.97 ± 1.45b	92.14 ± 1.26a	90.78 ± 1.90a	92.47 ± 2.58a	
Rhodospirillales	10.09 ± 2.63a	2.75 ± 0.57b	4.09 ± 1.15b	2.09 ± 1.37b	
Family					
Alcaligenaceae	1.47 ± 0.37a	0.98 ± 0.28bc	1.31 ± 0.13ab	0.55 ± 0.32c	
Bradyrhizobiaceae	63.20 ± 1.19b	86.43 ± 2.27a	81.71 ± 0.89a	87.25 ± 3.92a	
Geobacteraceae	1.48 ± 0.92a	1.22 ± 0.57a	1.65 ± 0.88a	0.80 ± 0.18a	
Cystobacterineae	0.16 ± 0.16a	0.05 ± 0.01a	0.06 ± 0.05a	0.05 ± 0.03a	
Nostocaceae	0.08 ± 0.01a	0.09 ± 0.03a	0.00 ± 0.01b	0.03 ± 0.02a	
Rhodospirillaceae	10.09 ± 2.63a	2.75 ± 0.57b	4.09 ± 1.15b	2.09 ± 1.37b	
Notes:

† Abbreviation for treatments are described in Table 1.

‡ Different letters within the same row indicate significant difference between treatments tested by one-way ANOVA (P < 0.05). Values are the means ± SE (n = 4).

At the order level, six groups, Burkholderiales, Desulfuromomadales, Myxococcales, Nostocales, Rhizobiales and Rhodospirillales were detected across all samples (Table 3). Among them, Rhizobiales was the dominant order, with an average relative abundance ranging from 81.97% to 92.47% across four treatments. Rhodospirillales was the second most abundant order, with average relative abundance ranging from 2.09% to 10.09%. The other orders had very lower abundances. Contrary to Rhizobiales, Rhodospirillales was significantly (P < 0.05) more abundant in the CC treatment.

At the family level, Bradyrhizobiaceae and Rhodospirillaceae were the two most abundant diazotrophs, and their average relative abundances ranged from 63.20% to 87.25% and from 2.09% to 10.09% across treatments, respectively (Table 3). Contrasting with Rhodospirillaceae, Bradyrhizobiaceae was significantly less abundance in the CC than in the other treatments.

Based on 97% similarity, only 51 different OTUs were obtained across all samples, indicating the composition of diazotrophic community was very simple. Among them, 15 OTUs had a relative abundance of more than 0.3% in at least one treatment (Table 4), and the distribution proportion of different OTUs into four treatments was illustrated in Fig. 3. Generally, OTU17 was a dominant member with an average relative abundance ranged from 63.01% to 87.26% across all treatments, and its relative abundance was significantly lower in the CC treatment than in the other three treatments. In the rotation system, OTU17 was lower in the CSC treatment than in the SCS treatment. A BLAST search at the amino acid level showed that OTU17 had 100% identity with Bradyrhizobium japonicum (ABO27443). OTU13 was the second most abundant member of the diazotroph and had 99% identity with Bradyrhizobium sp. (AKN21127). The abundance of OTU13 was at the highest level in CC than in the other treatments. Similarly, other more abundant OTUs were significantly higher in CC than in the other treatments. Particularly, several OTUs classified into the genus Skermanella (OTU5, 10, 26, 30, 31), which showed the highest abundance in the CC treatment (Table 4).

Table 4 Identification of the abundant OTUs (relative abundance > 0.3% at least in one treatment) at the amino acid sequence level by BLASTp on the NCBI website and changes in their relative abundances (%) as influenced by different cropping treatments.

OTU ID	Closest relatives	Access number	Identity (%)	CC†	SS†	CSC†	SCS†	
OTU1	Geobacter pickeringii	WP039743917	99	0.56 ± 0.46a‡	0.19 ± 0.03a	0.28 ± 0.14a	0.15 ± 0.11a	
OTU5	Skermanella aerolata	WP044431865	99	2.06 ± 0.75a	0.43 ± 0.06b	0.77 ± 0.11b	0.41 ± 0.32b	
OTU6	Burkholderiales bacterium JOSHI_001	WP009549047	100	4.74 ± 3.06a	0.44 ± 0.06b	1.75 ± 0.97b	0.40 ± 0.16b	
OTU7	Azospirillum	WP085088510	97	0.40 ± 0.23a	0.15 ± 0.04b	0.23 ± 0.04b	0.10 ± 0.07b	
OTU9	Uncultured bacterium	AHN51493	100	1.42 ± 1.14a	0.34 ± 0.14b	0.85 ± 0.48ab	0.35 ± 0.17b	
OTU10	Skermanella stibiiresistens	WP037454648	99	2.05 ± 0.64a	0.64 ± 0.17b	1.00 ± 0.32b	0.42 ± 0.24b	
OTU13	Bradyrhizobium sp. TUTMCJ4B	AKN21127	99	12.86 ± 3.04a	4.98 ± 1.30b	7.19 ± 2.08b	4.77 ± 3.02b	
OTU14	Burkholderiales bacterium JOSHI_001	WP009549047	99	1.20 ± 0.22a	0.71 ± 0.34b	0.70 ± 0.39b	0.42 ± 0.22b	
OTU16	Azohydromonas lata	WP084268130	99	0.70 ± 0.15a	0.40 ± 0.13bc	0.50 ± 0.26ab	0.16 ± 0.09c	
OTU17	Bradyrhizobium japonicum	ABO27443	100	63.01 ± 1.22c	86.59 ± 2.22a	81.73 ± 0.94b	87.26 ± 4.10a	
OTU26	Skermanella stibiiresistens	WP037454648	99	0.34 ± 0.06a	0.10 ± 0.05b	0.15 ± 0.09b	0.07 ± 0.07b	
OTU30	Skermanella stibiiresistens	WP037454648	97	2.30 ± 0.69a	0.68 ± 0.14b	0.97 ± 0.30b	0.48 ± 0.30b	
OTU31	Skermanella stibiiresistens	WP037454648	99	3.41 ± 0.87a	0.83 ± 0.17b	1.31 ± 0.37b	0.68 ± 0.49b	
OTU32	Burkholderiales bacterium JOSHI_001	WP009549047	99	0.42 ± 0.10a	0.06 ± 0.02b	0.07 ± 0.04b	0.03 ± 0.02b	
OTU41	Geobacter metallireducens	WP004514270	99	0.68 ± 0.64a	0.22 ± 0.12a	0.61 ± 0.83a	0.16 ± 0.07a	
Notes:

† Abbreviation for treatments as described in Table 1.

‡ Different letters within the same row indicate significant difference systems samples tested by one-way ANOVA (P < 0.05). Values are the means ± SE (n = 4).

Figure 3 Circular representation of microbial communities in CC, SS, CSS, SCS at OTU level.

The inner circular diagram show the relative abundance of different OTUs in different treatments. OTUs with relative abundance lower than 0.3% in all samples were not shown. The abbreviations of CC, SS, CSC and SCS are described in Fig. 1.

Changes of diazotrophic community structures

The PCoA plot of all diazotrophic community structures is shown in Fig. 4A. PCoA1 and PCoA2 explained 80.87% and 8.40% of the variation of the community structures, respectively, indicating that the diazotrophic community structures mainly changed along the PCoA1 axis. All samples were clearly separated into two main groups: one group contained samples of CC, and the other group consisted of samples from SS, CSC and SCS. Noticeably, although the diazotrophic communities among the treatments of SS, CSC and SCS were grouped closely with each other, the Adonis analysis indicated that the communities were significantly different between SS and CSC (P = 0.028), between CSC and SCS (P = 0.035), but not significant different between SS and SCS (P = 0.055) (Table S1). The envfit analysis showed that three soil factors were significantly correlated with change of community structures (Table S2). Of these, NO3−-N was positively correlated with the CC treatment, and AP was positively correlated with the SS, SCS and CSC treatments (Fig. 4B). In addition, the Mantel test showed that AP (r = 0.26, P = 0.038) was the most important factor in shifting the diazotrophic community structures (Table S3).

Figure 4 Principal coordinate analysis (A) and redundancy analysis (B) of soil diazotrophic communities in different cropped systems.

The abbreviations of CC, SS, CSC and SCS are described in Fig. 1.

Discussion

Changes in soil chemical properties in different cropping systems

Previous studies have shown that different cropping systems significantly change the soil chemical properties (Liu et al., 2017). For example, Jagadamma et al. (2008) and Miao, Qiao & Hang (2007) found that soil pH, TN, TC and available nutrient contents were significantly increased after long-term rotation of soybean. However, in this study, we found that soil pH and TC had no significant changes among the four treatments. No variation of soil pH observed in this study might be related to sampling time. The soils were collected near crop maturity in this study, and the soil pH as influenced by crop exudates might be very weak at this time, similar results was also reported by Meriles et al. (2009), who stated that the soil pH was significantly changed between continuous cropping soybean and soybean-corn rotation system at planting time, while no change was observed at harvest time. The finding of no significant change in soil TC content in this study is consistent with previous studies (Spargo et al., 2008; Liu et al., 2017), and may be related to the short-term experimental period (only 5 years) in this study. However, compared with the continuous cropping treatments of SS and CC, TN content in the rotation was significantly higher under the same crop (Table 1). The reasons for this may be related to two aspects: one is the rotation benefit for the growth of soybeans, which can fix more nitrogen from the air and promote the increase of TN content in soils (Adeboye, Iwuafor & Agbenin, 2006), and the other is that rotations including corn put more N fertilizer into the soil compared with the SS treatment, which can increase TN content in soil. Similarly, the changes in available nutrients, such as AP, NH4+-N and NO3−-N, may also be related to different fertilizer regimes between corn cropping and soybean cropping.

Effects of cropping system on the abundance of the nifH gene and diazotroph diversity

Previous studies have shown that different cropping systems significantly change the abundance of the nifH gene in agricultural soils (Reardon, Gollany & Wuest, 2014). In this study, we found that the abundance of the nifH gene was higher in cropping rotation than in continuous cropping when the crop was same (CSC vs CC; SCS vs SS) , and it was also higher when the growth crop was soybean than when the crop was corn (SS vs CC; SCS vs CSC) (Fig. 1A), suggesting that the abundance of the nifH gene is influenced by both cropping system and crop type. In addition, the finding that the nifH gene abundance was negatively correlated with NO3−-N in this study (Fig. 1C) is consistent with the result reported by Mirza et al. (2014), who stated that the increase in NO3−-N content in soil satisfied the demand of microorganisms for nitrogen, and the abundance of diazotrophs containing the nifH gene was significantly decreased. The finding that the nifH gene abundance is also negatively correlated with C/N (Fig. 1B) is supported by the finding of Wang et al. (2012), who stated the abundance of nifH gene in both rhizosphere and bulk paddy soil was negatively correlated with C/N under different period of organic management. Furthermore, the abundance of the nifH gene is positively correlated with AP (Fig. 1D), suggesting that AP is an important constraint in adjusting the ability of N2 fixation. Similar results were also observed in our previous study, we observed that the nifH gene abundance in a black soil treated with long-term of chemical fertilizers was positively correlated with available P content but negatively correlated with NO3−-N content (Hu et al., 2019).

The soil microbial community structures and the diversity index were affected by different cropping systems (Yin et al., 2010). Based on the meta-analysis of the soil microbial diversity index in different cropping systems, Venter, Jacobs & Hawkins (2016) stated that only 15.1% and 3.4% of the microbial community richness and diversity index in the rotation system showed an increasing trend, and the changes in the soil microbial diversity index were closely related to the different rotation systems and the years of continuous cropping. In this study, the Shannon index of diazotrophs in the CC treatment was significantly higher than that in the other treatments (Fig. 2), indicating that continuous cropping of corn significantly increased the alpha diversity of the diazotrophic community. This phenomenon may be related to the significant increase in the number of OTU17 in cropping systems including soybean (Table 4), since the increase of a single bacterium can lead to decrease community evenness and Shannon diversity index (Ortiz-Burgos, 2016). Consistent with the results of Navarro et al. (2013), there was no significant difference in the diversity index between the SS and SCS treatments (Fig. 2). In addition, there was no significant difference in the Chao1 and OTU number index among the treatments, suggesting that the crop systems significantly changed the abundance rather than species of the diazotrophic community. Furthermore, the finding that the Shannon index of the diazotrophic community was significantly negatively correlated with AP content (Table 2) was consistent with the findings of Eisenhauer (2016) and Hu et al. (2018), which suggested that available P nutrient increased the nifH abundance but relatively simplified the diversity of diazotrophic community. Our findings strongly highlighted that the changes of the diazotrophic community is highly sensitive to P content, and P is an important element for microorganism growth and development (Jean et al., 2013).

Effects of cropping systems on diazotrophic community structures

The diazotrophs in the black soil ecosystem are dominated by Proteobacteria (Ding et al., 2016). Meanwhile, Pereira E Silva et al. (2013), based on PCR-DGGE analysis, has shown that Bradyrhizobium is the most abundant genus of diazotrophs in long-term rotation systems. In this study, we found that compared with the CC treatment, the abundance of Rhizobiales was significantly higher in the SS, SCS and CSC treatments (Table 3), which is mainly because most Rhizobiales are symbiotic microbes with the roots of legumes. Inderjit (2005) reported that the quantity and quality of crop secretions from different crops could lead to changes in diazotroph colonization. Consistent with the results of Pereira E Silva et al. (2013), this study found that B. japonicum is the dominant species of the diazotrophs in all treatments. We found that the most abundant OTU17, which was classified as B. japonicum, was significantly higher in SS, CSC and SCS than in CC (Table 4), suggesting that a cropping system including soybean is beneficial to the multiplication of this bacterium. Noticeably, although continuous cropping of corn was conducted for 5 years in this study, the members belonging to the Bradyrhizobium sp. (OTU17 and OTU13) were still in higher abundance than other OTUs in the CC treatment (Table 4). We observed that this result is not surprise since Bradyrhizobium sp. can act as an active endophytic plant growth promoter associated to plants beside of legume (Videira et al., 2013; Rouws et al., 2014; De Alencar Menezes Júnior et al., 2019). Recently, a study also showed that Bradyrhizobium sp. is the dominant member of the diazotrophic community in the annual rotation of summer corn with winter wheat in southern China (Wang et al., 2017a). One of the interesting findings is that the CC treatment led to a higher abundance of OTU13 than in the other three treatments (Table 4). This finding was not reported previously, and the reasons need to be revealed with further studies. Another noteworthy finding was that the OTUs classified into the Skermanella genus had the highest abundance in the CC treatment than the cropping system including soybean (Table 4). However, the relative abundance of Skermanella in CC was only about 10%, which significantly lower that than 50% of Skermanella observed in 35 years of monoculture of corn in a black soil (Hu et al., 2018). Given the monoculture cropping corn in this study is only 5 years, we speculate that the relative abundance of Skermanella will be increased with years of continuous corn cropping in the black soils. Although the abundance of Skermanella was increased in the continuous corn cropping system, this event is not benefit for increasing soil N nutrition, because this genus contains nifH gene but unable to fix N2 (Zhu et al., 2014).

The diazotrophic community structure in CC was significantly different from that in the other treatments (Fig. 4A). Meanwhile, the structure of diazotrophs in SCS was markedly different from that in the CSC treatment, suggesting that different crops significantly change the diazotrophic community structure (Kent & Triplett, 2002; Wardle et al., 2004). Inderjit (2005) reported that the main reason for forming different microbial communities is the allelopathic effects between plants and microorganisms. Soybean and corn secrete different root exudates into the soil, inducing different microbial colonization and a correspondingly change the diazotrophic community structure in soils. Interestingly, there was no significant difference in the diazotrophic community structures between the SS and SCS treatments, indicating that when the growing crop is soybean, the diazotrophic community structures are not influenced by the cropping system. However, given only 5 years of continuous soybean cropping involved in this study, whether long-term cropping, such as longer than 10 years of continuous soybean cropping, could change the diazotrophic community structures compared with SCS needs to be addressed in future studies.

Both RDA and the Mantel test results showed that AP had the greatest contribution to the changes of the diazotrophic community structures (Fig. 4B; Table S3), indicating that AP is the main driving factor for the variation of diazotrophic communities after changes in cropping systems. Our findings strongly highlight that the changes of the diazotrophic community in this study from aspects of abundance, diversity and community structure are highly sensitive to available P content. The finding of this study is lined with the several previous studies, which indicated that soil phosphorus content is a limiting factor for N2 fixation, and available P induced the largest impacts on soil diazotrophic community (Vitousek et al., 2002; Tang et al., 2017; Hu et al., 2018).

Conclusions

In summary, our study demonstrated that several soil properties such as AP, C/N and nitrogen related nutrients were significantly influenced by cropping systems. We found that the abundance of the nifH gene, diversity and structures of diazotrophic communities differed with cropping systems, especially in continuous corn cropping. The soil properties, specially of soil available P was tested as the critical factors in driving the changes of diazotrophic community, which highlighted that the management of soil phosphorus nutrient is very important for increasing BNF in black soils. In addition, we observed that the relative abundance of Bradyrhizobium sp. was significantly increased under the cropping systems including soybean, while the abundance of Skermanella sp. was increased under continuous corn cropping, which suggested that the cropping systems including soybean is favorable for BNF.

Supplemental Information

Supplemental Information 1 Raw data of nifH gene abundance and soil chemical properties displayed in Figure 1 and Table 1.

Click here for additional data file.

Supplemental Information 2 A schematic diagram showing the cropping sequences from the starting year 2013, to the sampling year 2017, among the four treatments.

CC and SS represent the treatments of continuous cropping of corn and soybean, respectively; CSC and SCS represent the treatments of soybean-corn rotation for growing corn and soybean, respectively.

Click here for additional data file.

Supplemental Information 3 Adonis analysis of the differences in diazotrophic community structures between treatments.

Click here for additional data file.

Supplemental Information 4 Envfit analysis of the correlations between the diazotrophic communities and soil factors.

Click here for additional data file.

Supplemental Information 5 Mantel test to determine the correlations between the diazotrophic communities and soil factors.

Click here for additional data file.

Additional Information and Declarations

Competing Interests

Author Contributions

Data Availability

The authors declare that they have no competing interests.

Jiaxun Zou conceived and designed the experiments, performed the experiments, analyzed the data, prepared figures and/or tables, authored or reviewed drafts of the paper, and approved the final draft.

Qin Yao conceived and designed the experiments, performed the experiments, prepared figures and/or tables, authored or reviewed drafts of the paper, and approved the final draft.

Junjie Liu analyzed the data, prepared figures and/or tables, and approved the final draft.

Yansheng Li performed the experiments, prepared figures and/or tables, and approved the final draft.

Fuqiang Song conceived and designed the experiments, authored or reviewed drafts of the paper, and approved the final draft.

Xiaobing Liu conceived and designed the experiments, authored or reviewed drafts of the paper, and approved the final draft.

Guanghua Wang conceived and designed the experiments, authored or reviewed drafts of the paper, and approved the final draft.

The following information was supplied regarding data availability:

Data is available at NCBI: PRJNA516581.

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
