# Peer review of "Changes of diazotrophic communities in response to cropping systems in a Mollisol of Northeast China"

_PeerJ, doi:10.7717/peerj.9550_

## Round 0.1 · original submission · Minor Revisions

All reviewers and myself have seen merit on your work and provide interesting results. They have provided useful comments and suggested modifications for improvement. I advise you to take all comments into account and send a revised version of the manuscript.

dReviewer 1 ·

Basic reporting

Introduction and background prepared to show context that the experience were developed.
Literature well referenced and relevant. It has some ancient references, but concerning the subject of study.
The figures are relevant, high quality, well labeled and described.

Experimental design

Methods described with sufficient detail and information to replicate.

Validity of the findings

All underlying data have been provided; they are robust, statistically sound, and controlled.
Conclusions are well stated, linked to original research question & limited to supporting results.

Additional comments

If it is possible, eliminate ancient bibliography, or part of it.

Reviewer 2 ·

Basic reporting

Clear and unambiguous, professional English used throughout.
Literature references, sufficient field background/context provided
Professional article structure, figures, tables. Raw data shared.

Experimental design

Research question well defined, relevant & meaningful. It is stated how research fills an identified knowledge gap.
Rigorous investigation performed to a high technical & ethical standard.

Validity of the findings

All underlying data have been provided;

Additional comments

1 a)]. Please, re-word and discuss accordingly.
20) Lines 283 – 285: Please, relate Shannon index and AP back to the cropping systems.
21) Line 292: Please, replace “breeding” with “yields”.
22) Line 294: In fact, Bradyrhizobium japonicum was, by far, the dominant species in all treatments (Table 4). Please, correct and discuss accordingly.
23) Lines 296 – 297: Please delete commas after systems and soybean.
24) Line 308: “… dominant diazotroph under 35 years of monoculture of corn in a black soil…” this is good information. It would be interesting to state why was Skermanella dominant under the mentioned conditions.
25) Line 328: “…previous report has shown that soil phosphatase activity was significantly increased in crop rotation…” If this is true, why SS (continuous cropping) and CSC (intercropping) were not significantly different in terms of available phosphorous (Table 1)? Please, correct and discuss accordingly.
26) Line 335: Please, relate your conclusions to the three objectives stated in the introduction.

Reference

1. Fehr, W.R. and Caviness, C.E., Stages of soybean development. 1977: Iowa State University of Science and Technology Ames, Iowa.

Annotated reviews are not available for download in order to protect the identity of reviewers who chose to remain anonymous.

Reviewer 3 ·

Basic reporting

Manuscript # 44497
Title: Changes of diazotrophic communities in response to cropping systems in a Mollisol of Northeast China
Authors: Jiaxun Zou, Qin Yao, Junjie Liu, Yansheng Li, Fuqiang Song, Xiaobing Liu, Guanghua Wang

In this study, the authors examined diazotrophic microbial communities in different cropping systems. The study, overall, is well designed and has strong potential in expanding our understanding of how these specific microbial communities compare in different cropping systems. The manuscript is very well written - clear, professional and articulate English has been used throughout. Authors have used good literature citations throughout. Sufficient background, rationale and discussions for the work has been included in the manuscript. The figures and tables look good and support the study and the collected dataset well.

I have the following comments about the manuscript, please see below.

Lines 68-69: needs a few references after this line.

Line 73: I suggest you write “rarely specifically related to the diazotrophic community”

Section 2.1: Study site descriptions and soil sampling - Were samples taken only once? Authors need to include this information.

Section 2.3:
• Line 119-127- qPCR conditions – Authors need to include more detail on how the qPCR standards were generated and used: e.g., What was exactly used as a standard? What exactly is “known number”? What concentrations and range of standards were used? Was there an internal control used in the study? What other controls were used for your qPCR reactions?

• Line 123: Instead of volume of primers, please provide the final primer concentrations in your final mastermix volume. Volume of primer really does not inform much if someone would like to refer to your manuscript in future.

Line 265: Please elaborate on the levels of C and N here (in your reference article) and how exactly that compare to this particular study.

Line 267-268: Please elaborate in a sentence the exact findings from this reference and how specifically this compares with your data.

Figure 3: From what I gather, this tree has been constructed with sequences across all samples. If possible, I suggest you make this separate: make one neighbor joining tree each for each treatment. That comparison will be highly informative for future references. If space doesn’t permit, you can also include the three separate trees as supplemental information.

Table 3: I suggest you make/add a plot of this instead of the table. On sequence analysis platforms such as QIIME, you should be able to generate stacked plots of each of the diazotrophic bacterial groups based on each of the cropping systems. That certainly will be more informative and serves as a better way to display this particular result.

Experimental design

See all my comments above

Validity of the findings

See all my comments above

·

Basic reporting

In the manuscript “Changes of diazotrophic communities in response to cropping systems in a Mollisol of Northeast China” Zou and colleagues assessed the abundance of nifH gene and their sequences in and Mollisol in China under different soybean/corn management. They found that the crop rotation changes the abundance and composition of diazotrophic community in the soil. It is not a great scientific novelty, but they prove their hypothesis applying a very robust and high-throughput approach.

Experimental design

Despite to be a very new experiment, only 5 years to crop rotation, they found interesting results showing that, even after a short time, the diazotrophic communities change dus to the crop management.
The molecular and statistical analysis were well done.
Some comments are given in the section “General comments” to improve he quality of your materials and methods description.

Validity of the findings

The findings support the hypothesis that the crop rotation changes the diazotrophic communities in agricultural soils and brings new results to Mollisols. Some comments are given to improve your discussion.
Your conclusions must be rewritten according to stated in the next section.

Additional comments

Introduction:
Your introduction is very straight. Congratulations.
But I suggest to add more citations, and proper comments, of studies with diazotrophs in your Mollisols, even those based in isolation and diversity assessment (if there are some paper).
Materials and methods
Line 85: The average temperature in your experimental site is 1.5°C?? Are you sure?
Lines 87-89: State that your treatments CSC and SCS have annual rotation of crops and that, at the moment of sampling you had, corn and soybean, respectively, in the sites.
Lines 105-112: Please, cite the methods for each soil analysis.
Lines 126-127: It is not clear how do you prepared you standard curve. In the paper referenced (Yao et al., 2017), this information is also not clear. Please, state it clearly. Did you used a cloned nifH gene? Or a known DNA concentration of a given diazotrophic bacteria?
Results and Discussion
Your results are very well presented. Here, I made some appointments in the results for the further discussion.
Lines 174-180: The highest abundance of nifH copies in the cropping sites where soybean were grown in the sampling time (SCS and SS) is expected for two reasons. 1) the increasing of indigenous Bradyrhizobium populations and 2) the less N fertilizers applied. You should address some comment about this fact in the discussion of this topic, increasing your discussion in the lines 255-268.
Table 3: Please, take a look in the letters of your average comparison test. Sometimes the “a” is higher and sometimes is lower.
Lines 212-222: As well as commented to the qPCR analysis, the abundance of Bradyrhizobium in soybean growing sites is expected. Please, address this discussion in the proper section.
Lines 224-235: Note that your differences in Adonis analysis is related to the crop in the m moment of the sampling (soybean not equal to corn).
Line 241: Do you think that 5 years of crop management is enough to change such stable soil features such as pH and Carbon? You answered below!
Lines 297-305: Some papers sowed that Bradyrhizobium is an active endophytic plant growth promoter associated to grasses, for example sugarcane and elephant-grass. May be in your corn fields this populations were increased due to the active colonization of the corn plants.
The papers above should you include this discussion in this point.
Videira et al. (2013) Culture-independent molecular approaches reveal a mostly unknown high diversity of active nitrogen-fixing bacteria associated with Pennisetum purpureum — a bioenergy crop. Plant Soil 373:737–754.
Rouws et al. (2014) Endophytic Bradyrhizobium spp. isolates from sugarcane obtained through different culture strategies. Environ Microbiol Rep 6:354–363.
Menezes Júnior et al. (2019) Occurrence of diverse Bradyrhizobium spp. in roots and rhizospheres of two commercial Brazilian sugarcane cultivars. Brazilian J Microbiol. 50: 759-767.
Conclusions:
You are repeating some results. I edited your text to try a better conclusion. Please, add some text straight like this.
“The diazotrophs abundance and community structures differed with cropping systems especially continuous corn cropping. The soil parameters played a critical role in shaping the diazotrophic communities, specially to the available phosphorus. The continuous cropping and rotation systems significantly altered the abundance, diversity and community structures of the diazotrophic community in black soils”.

---

## Round 0.2 · Minor Revisions

Thanks for taking into consideration most of the reviewers' comments, I believe your manuscript has improved a lot. However, there are still some important suggestions and recommendations made by two reviewers that need to be taken into account.

Reviewer 2 ·

Basic reporting

The manuscript was most times clear but requires further editing before it be recommended for publishing. Important and generally updated references are used but a few other references are required to substantiate key findings. Tables and figures are generally well presented but key editing is required so that the major audience of this is journal can understand the article.

Experimental design

The knowledge gap is well identified. The study is conducted with high technical standard and the methods are clearly explained.

Validity of the findings

The findings are valid. The key information has been provided. Conclusions are well stated and linked to the original research questions.

Additional comments

Changes of diazotrophic communities in response to cropping systems in a Mollisol of Northeast China

General Comment

This is the second review of this manuscript. The general suggestion is that the authors should now focus on improving the discussion section organizing it in paragraphs rather than subsections. They should not discuss every single result but the key findings, considering the objectives of the study. Suggestions on how to improve the presentation of tables is also provided. English writing should be improved in all sections. The specific recommendations are listed below.

Specific Comments

In Table 1: please should indicate mean ± standard errors. CC, SS, CSC and SCS meaning should be either provided in the title or with different symbols. As the symbol appear now “a” represents cropping systems and the statistically best treatment mean. The same advice applies for soil properties (TN, TC ….etc.). Make sure to provide a legend for the statistical mean-ranking symbols below the table.
Table 3: Please, include (%) in the title after abundance.
Line 40: “Although the barriers of continuous corn cropping are not as serious as those of continuous soybean”. The meaning is not clear.
Line 50: “Approximately 110 Tg of N is input into terrestrial ecosystems annually through the BNF method (Gruber et al., 2008)”. It is not a good idea to use over ten years old estimates, especially when more recent calculations, for example Davies‐Barnard and Friedlingstein (2020), are available.
Line 91: please, change text so that it reads: “Commercial fertilizers urea, diammonium…”

Line 178: Please, change text so that it reads “The abundance of the nifH gene in the SCS treatment was 57.4%, 123.8% and 382.3% higher compared …”

Lines 179 – 181: Please, change text so that it reads “Pearson’s correlation analysis showed that nifH gene abundance was significantly correlated with C/N (r = -0.541, P = 0.031), NO3--N (r = -0.602, P = 0.014), and AP (r = 0.597, P = 0.015) (Fig. 1B, C, D).”

Lines 183 – 184: Please, change text so that it reads “Shannon and Simpson indices had opposite results. CC treatment had the highest score with Shannon index but had the lowest score with Simpson index.”
Line 193: Please, replace “99.08” with “99.91”.
Lines 194 – 195: Please, change text so that it reads “… while the highest relative abundances of Betaproteobacteria and Deltaproteobacteria were only 1,37% and 1.71%, respectively.”

Line 198: Please, help your readers by listing the orders either alphabetically or in the same sequence they appear in the table.
Line 202: Please, change text so that it reads “Contrary to Rhizobiales, Rhodospirillales was significantly (P < 0.05) more abundant in the CC treatment.”
Line 207 – 209: Please, change text so that it reads “Contrasting with Rhodospirillaceae, Bradyrhizobiaceae was significantly less abundance in the CC than in the other treatments.”

Lines 242 – 243. “No variation of soil pH observed in this study might be ascribed into two reasons: one is the soil samples were collected from bulk soil not of rhizosphere soil which has more buffer effect against the changes of soil pH, another might be related… ” Jagadamma et al. (2008), authors that you cited, also sampled bulk soil but observed differences in soil pH. I would delete bulk soil as one of the reasons.
Lines 249: Please, replace “which” with “and”.
Lines 251”Please, replace “increased” with “higher”.
Lines 254 – 255: “ … rotations including corn put more N fertilizer into the soil compared with the SS treatment, which can increase TN content in soil…”. Please, substantiate this affirmation with relevant and recent literature.
Line 256: “…may also be related to different fertilizer regimes between corn cropping and soybean cropping”. Please, elaborate.
Line 281: “This phenomenon may be related to the significant increase …” Please, make sure this is consistent with the previous statement.
Line 295: “Bradyrhizobium is the most abundant genus of diazotrophs in long-term rotation systems”. This if a very important information. Please, provide an explanation, the possible reason for this.
Line 297: Please, replace “increased” “with “higher”.

Reference
Davies‐Barnard T, Friedlingstein P (2020). The global distribution of biological nitrogen fixation in terrestrial natural ecosystems. Global Biogeochemical Cycles 34 (3): e2019GB006387. DOI: 10.1029/2019GB006387

Annotated reviews are not available for download in order to protect the identity of reviewers who chose to remain anonymous.

Reviewer 3 ·

Basic reporting

The authors have made sufficient changes to the manuscript.

Experimental design

no comment

Validity of the findings

no comment

Additional comments

no comment

·

Basic reporting

The manuscript written by Zou el al. reports the changes in the diazotrophic soil communities due to different crop/soil management. In my first review, the paper showed several gaps that should be filled to increase the paper quality and reach those needed to be published in PeerJ. Now, the quality of the paper increased a lot. I congratulate the authors for their commitment to the reviewer's comments and to the effort to increase the paper quality.
A single point. In line 363, remove the word “many”, since the paper cited studied only two non-legume crops (elephant-grass and sugar-cane).

Experimental design

The quality of experimental presentation increased a lot.

Validity of the findings

The data presentation and discussion is more clear now.

Additional comments

The paper can be accepted for publication.

---

## Round 0.3 · accepted · Accept

Dear authors, your manuscript improved after the revision you made including all comments from both rounds of revision. Now your manuscript is ready to be published.